# ISNS General Guidelines for Neonatal Bloodspot Screening 2025

**DOI:** 10.3390/ijns11020045

**Published:** 2025-06-14

**Authors:** Dianne Webster, Amy Gaviglio, Aysha Habib Khan, Mei Baker, David Cheillan, Layachi Chabraoui, Ghassan Abdoh, Juan Cabello, Roberto Giugliani, Dimitris Platis, Jan Østrup, R. Rodney Howell, Peter C. J. I. Schielen, James R. Bonham

**Affiliations:** 1National Newborn Screening Laboratory, LabPlus, Health New Zealand Te Whatu Ora, Auckland 1023, New Zealand; diannew@adhb.govt.nz; 2Connetics Consulting, LLC, Minneapolis, MN 55417, USA; amy.gaviglio@outlook.com; 3Department of Pathology and Lab Medicine, Aga Khan University, Karachi 74800, Pakistan; aysha.habib@aku.edu; 4Department of Medicine, Aga Khan University, Karachi 74800, Pakistan; 5Wisconsin State Laboratory of Hygiene, School of Medicine and Public Health, University of Wisconsin-Madison, Madison, WI 53726, USA; mei.baker@slh.wisc.edu; 6Centre Régional de Dépistage Néonatal Auvergne Rhône-Alpes (CRDN AuRA), 69003 Lyon, France; david.cheillan@chu-lyon.fr; 7Faculty of Medicine and Pharmacy, University Mohammed V of Rabat, Rabat 10100, Morocco; lchabraoui@yahoo.fr; 8Newborn Screening/Neonatal Intensive Care Unit, Pediatrics Department, Hamad Medical Corporation, Doha P.O. Box 3050, Qatar; ghassanabdoh@yahoo.com; 9Instituto de Nutrición y Tecnología de los Alimentos (INTA), University of Chile, Santiago 7830489, Chile; jfcabello@gmail.com; 10Postgraduate Program in Genetics and Molecular Biology UFRGS, Medical Genetics Service HCPA, Dasa Genomica, and Casa dos Raros, Porto Alegre 90610-261, Brazil; rgiugliani@hcpa.edu.br; 11Department of Newborn Screening, Institute of Child Health, 11527 Athens, Greece; dplatis@ich.gr; 12International Society for Neonatal Screening, Reigerskamp 273, 3607 HP Maarssen, The Netherlands; ostrup.jan@gmail.com (J.Ø.); j.bonham@nhs.net (J.R.B.); 13Hussman Institute for Human Genomics, Miller School of Medicine, University of Miami, Miami, FL 33136, USA; rhowell@med.miami.edu

**Keywords:** biobanking, parent consent, laboratory practice, quality assurance, case definition, screening system, data access, auditing, screening pathway, patient referral, screening panel, ELSI considerations

## Abstract

Part of the vision of the ISNS is ‘to enhance the quality of neonatal screening and medical services through dissemination of information, guidelines and best practices.’ Although newborn screening encompasses testing in the newborn period for critical congenital heart disease, hearing impairment, birth defects, and congenital biochemical disorders (usually on bloodspots), this guideline is specifically about bloodspot screening. The ISNS has provided neonatal screening guidelines for many years and here presents the renewed 2025 General Guidelines for Neonatal Bloodspot Screening. They are intended to provide a framework for screening programs to develop specific policies around all aspects of the newborn screening system, offering the basic set of items for consideration. These guidelines provide trusted anchors to build, expand, or maintain robustly organized neonatal or newborn screening (NBS) programs and a checklist to evaluate and improve the essential elements of those programs. For starting or developing programs, it is a set of elements for which provisions need to be in place and a checklist of items that the screening program should at a minimum have provisions for. The publication of these guidelines is meant as a starting point for interactive discussion, to further improve this document and expand where necessary.

## 1. Introduction

Neonatal screening or newborn screening (NBS) is a public health activity intended for serious treatable conditions shortly after birth. It has been recognized as an effective public health intervention by the WHO [1]. NBS is historically a successful public health program; many countries in the world have well-organized screening programs, and an estimated 38–40 million babies are screened each year, right after birth, for at least one or more disabling disorders (or conditions or diseases—we will use disorder throughout this document). Early detection and treatment enables the achievement of maximum health and development potential in affected infants.

Like other population screening programs, NBS results in harms as well as benefits, and programs must always aim to offer more benefit than harm (at a reasonable cost). Many, but not all, programs are built taking into account sets of ethical and practical rules to maintain programs that maximize health gain and avoid harm to the screened population. Not all developed programs have such rules, and they are an essential part of introducing a new program. This prompted the ISNS to provide anchors for ethically and practically sound neonatal screening.

The vision of the ISNS includes ‘the enhancement of the quality of neonatal screening and medical services through dissemination of information, guidelines and best practices.’

The ISNS has provided neonatal screening guidelines for many years and here presents the renewed 2025 General Guidelines for Neonatal Bloodspot Screening.

They are intended to provide a framework for screening programs to develop specific policies around all aspects of the newborn screening system, offering the basic set of items for consideration. They can provide trusted anchors to build or maintain robustly organized NBS programs when the dynamics of innovation may easily disturb the fabric of well-organized screening programs. For developing programs, it is a set of norm elements for which provisions need to be in place. It can serve as a checklist of items for which the screening program should have appropriate provisions.

The initiative to update the guidelines was under the direction of the ISNS Council. It followed a staged, collaborative, consensus-based approach involving expert contributions and iterative revisions. First, a strategic process was agreed upon to engage the Executive Board of the ISNS (a subset of the ISNS council), followed by a Council review and broader consultation with members and affiliated organizations, including sister societies such as the Association of Public Health Laboratories (APHL).

While colleagues have been consulted, the consultation of all stakeholders is impossible. Therefore, these guidelines are published by the society (ISNS) with the current council members as authors. This will ensure broad access and implementation across global newborn screening communities.

The publication of these guidelines at this time offers a starting point for interactive discussion. All stakeholders are invited to critically review the Guidelines and identify points for improvement. Please send your comments to the corresponding author of this document or, alternatively, send a letter to the Editors of *IJNS*. (https://www.mdpi.com/journal/IJNS/about, accessed on 9 June 2025). Valid points will be added to a subsequent update of these guidelines.

## 2. Terminology

For terminology in neonatal screening, we refer to the Clinical and Laboratory Standards Harmonised Terminology Database https://htd.clsi.org/listalltermsNewborn.asp, accessed on 9 June 2025). 

In particular, the following definitions from CLSI are used in this document, noting that neonatal screening is used in place of newborn screening:

**Newborn screening program**—A health program, which is one part of a larger newborn screening system, that operates with the goal of reducing morbidity and mortality in newborns with the biochemical or genetic screening of congenital disorders through early detection and intervention and consists of the jurisdiction’s health service components, which might include policies and regulations, planning and audits, specimen collection and transport, laboratory testing, and short- and long-term follow-up.

**Newborn screening system**—A collaboration of newborn screening stakeholders, including public and private agencies, organizations, families, policy makers, healthcare providers, and other caregivers, working together to ensure that all newborns within a defined geographical area have access to newborn screening and that babies found affected are able to access appropriate care and optimize health outcomes.

**Positive screen**—A final, reportable result for a disorder, group of disorders, or phenotypic difference, based on the newborn screening result(s) and laboratory screening algorithm, indicating that the risk for that disorder, group of disorders, or phenotypic difference is higher and that additional follow-up is needed.

**Short-term follow-up**—Steps taken to ensure a final screening outcome for newborns with actionable screening results.

Additionally, we refer to the screening pathway, the series of events which occur for a baby—for bloodspot screening, this includes the provision of information to families, the offer of screening, collecting the sample, transport to the laboratory, testing, reporting results, and, if necessary, diagnostic workup and treatment.

## 3. Framework for Neonatal Screening

The ISNS considers the Wilson and Jungner principles as the framework for responsible neonatal screening. This framework has most recently been re-iterated in this 2020 document of the WHO’s European branch [2].

The Wilson and Jungner principles are the following:The condition should be an important health problem.There should be an accepted treatment for patients with recognized disease.Facilities for diagnosis and treatment should be available.There should be a recognizable latent or early symptomatic phase.There should be a suitable test or examination.The test should be acceptable to the population.The natural history of the condition, including development from latent to declared disease, should be adequately understood.There should be an agreed policy on whom to treat as patients.The cost of case-finding (including a diagnosis and treatment of patients diagnosed) should be economically balanced in relation to possible expenditure on medical care as a whole.Case-finding should be a continuous process and not a “once and for all” project.

### 3.1. Treatment

Accepted treatment (Principle 2) should be proven to result in reduced morbidity and mortality for the baby screened; the secondary consequence of this may also benefit the baby and the wider community. Treatment in the current neonatal screening context is considered to be under medical management and, in addition to medications and dietary modification, may include advice or continued medical surveillance where a diagnosis is uncertain. For the assessment of this criterion, treatment does not include informing future reproductive choices or shortening the diagnostic odyssey.

### 3.2. Disorder Definition

Principles 2, 7, and 8 refer to ‘recognized disease’, ‘the natural history of the condition’, and ‘agreed policy on whom to treat as patients’. The disorders on neonatal screening panels present as a spectrum of severity determined by multiple genetic and environmental factors. The extent of the spectrum is frequently not known before population screening as the same biochemical and genetic markers of clinically presenting disease may cause mild or no symptoms and hence be unrecognized. Any available international definitions should be used, and without such a definition, the program must define the target disorder (e.g., infantile onset, or ‘salt-wasting’, by clinical evaluation or diagnostic findings).

### 3.3. Selection of Disorders for Screening Panels

To summarize the Wilson and Junger criteria, screening is recommended for disorders where there is a demonstrated benefit from early diagnosis and effective treatment, the benefit is balanced against financial and other costs, there are acceptable tests, and follow-up services are available and accessible for management.

Screening tests should not be recommended if indications of advantage from early diagnosis and treatment are lacking or uncertain, or the test is unsuitable.

In situations where a disorder may be considered a credible candidate for screening but evidence is lacking, it may be possible to gain this evidence from a carefully designed and time-limited evaluation. In these circumstances, this should be conducted and the findings published.

There are often disorders that may be detected as an incidental finding when screening for a recommended disorder. These should be reported (from the laboratory or following diagnostic testing), although it is recognized that the screening pathway may not be optimized to ensure their detection.

The neonatal screening system should have a pathway for the consideration of disorders for addition to screening panels.

Note that when it is possible to detect more than one disorder from a single (multiplexed) test, the disorders may be considered individually or as a package (e.g., if an additional disorder can be detected at minimal cost with high specificity screening, including the disorder may be approved even if sensitivity is limited).

## 4. The Screening Pathway

The screening pathway consists of the screening offer, sample collection and transport to the laboratory, sample processing and laboratory analysis (and, if applicable, biobanking), the reporting of results, referral, confirmatory (diagnostic) testing, and treatment.

Guidance for healthcare professionals supporting screening should be available as a special document or as described in regulations or recommendations from the governing body.

### 4.1. Offer of Screening

The program must have a policy covering pre-screening information, the screening offer and consent or refusal, considering local custom and practice, and any mandate and action when screening is declined.

Information about screening should be offered in a culturally appropriate way, ideally in the antenatal period. It should contain information about the screening process, the screened disorders, how results will be communicated, and general program information (such as sample storage and use). For more information on parent information, see (https://www.isns-neoscreening.org/wp-content/uploads/2021/04/ISNS-infographic.pdf, accessed on 12 June 2025). 

Note: Consent to screening has two aspects—to the screening process and to the procedure of taking the bloodspots.

Where this is required by the jurisdiction, family consent to screening should be obtained and recorded according to local healthcare practice.

### 4.2. Specimen Collection and Transport to the Laboratory

#### 4.2.1. Time to Specimen Collection

The guidance for healthcare professionals should address the recommended time for specimen collection and means of transport to the laboratory, taking into account local maternity practices and the competing needs of different disorders and the temporal evolution of marker levels.

#### 4.2.2. Trained Professionals Perform Specimen Collection

The specimen collection of capillary blood via heel stick or other means should be performed by trained personnel according to local policy. Comprehensive information on dried bloodspot specimen collection for neonatal screening can be found in CLSI document NBS01 (https://clsi.org/media/t1sdo14z/nbs01_sample.pdf, accessed on 9 June 2025). In summary, a validated collection paper must be used, as well as any demographic information necessary for result interpretation and reporting collected.

#### 4.2.3. Transport of Bloodspots

Bloodspots must be transported in a safe manner according to local regulations. Transport must be as rapid as feasible and protect the specimens from extremes of temperature and humidity.

### 4.3. Specimen Processing and Laboratory Analysis

#### 4.3.1. Laboratory Accreditation

Screening laboratories must work to the same standards as other clinical laboratories in the jurisdiction. Ideally, they should be accredited by a reputable agency against national or international standards (e.g., those in the Clinical Laboratory Improvement Act (CLIA) or ISO15189). Regular audit ensures testing is safe and reliable and the laboratory has suitable quality assurance processes in place, including participation in external quality assurance programs. A list will be available on the ISNS website.

#### 4.3.2. Laboratory Size

The appropriate number of specimens for a laboratory is dependent on geography, the tests performed, and organizational and economic considerations. Smaller laboratories may be needed in isolated regions or archipelagoes. A small number of specimens may be appropriate if cost-effective technology is available and the tests have stringent internal and external quality control.

Contingency planning should be in place and thoroughly tested to ensure continuity of service through natural and other disasters affecting maternity and laboratory service personnel, sample transport, and laboratory and hospital facilities. The laboratory aspects may be covered by an accreditation process, but all other steps of the screening pathway should be considered.

### 4.4. Specific Disorder Guidance

The Clinical and Laboratory Standards Institute has an extensive catalog of guidelines for screening specific disorders. Guidelines have been developed with ISNS input and are recommended as a comprehensive source of detailed information and advice (https://clsi.org/media/w1ancqx1/catalog2022_web.pdf, pg 28–29; accessed on 9 June 2025).

For most screened disorders, screening specificity can be improved through the addition of sequential (second-tier) testing on specimens with out-of-range levels of the primary marker metabolite. Where it is feasible, the testing of additional metabolites and/or the use of metabolite ratios and/or genomic testing will improve screening and provide additional information for the family and treating physician.

### 4.5. Reporting Results

#### 4.5.1. Unsuitable Specimens or Missed Specimens (NBS01)

A provision should be in place to identify specimens not arriving at the laboratory and identifying specimens that are unsuitable for analysis or may produce faulty results. In those cases, a repeat specimen should be arranged as soon as possible and analyzed. The laboratory should have a policy regarding testing unsuitable specimens or not and reporting significant results if such specimens are tested.

#### 4.5.2. Referral of a Positive Screening Result/Short-Term Follow-Up

For each screened disorder, the laboratory should have a reporting protocol that may include different communication when urgent action is required. Clinical critical results should be reported to a person who will take responsibility for further necessary action, and the laboratory should be confident that the person is accepting responsibility for follow-up.

Results indicating a screened disorder is probable should be communicated to families by someone familiar with the disorder who can ascertain whether urgent medical attention is required and outline the next steps to the family. This may be supported by just-in-time written information for family reference and sharing with the wider family.

Genetic counseling should be provided when it is appropriate for the disorder at a time suitable for the family.

Timely confirmatory (diagnostic) testing reflecting the case definitions is essential. Protocols should be established and consistently applied with a short and defined turnaround time to allay parental anxiety and stress and maximize health outcomes in affected newborns.

Screening test results should ideally be part of the baby’s health record. Where testing includes genomic results, particular care needs to be taken with regard to data privacy and security. These results should be part of the child’s health record to enable the regular curation of variants and access when there is clinical need.

Where this is feasible, families should receive negative screening test results (i.e., test results not eliciting referral).

## 5. Neonatal Screening System

Neonatal screening systems have multiple components that must work together to obtain optimal outcomes for screened babies. The responsibilities of each component or group of components can be defined, and policies can be developed that apply to the program.

One or more components of the system (usually the funding body) will be responsible for program governance. The system may operate at a national, regional, or local level, with responsibility often taken by a ministry of (public) health or similar body.The governance arm of the program will work in a wider context (for example, with other government ministries) to provide appropriate legislation, regulations, or recommendations to protect the program.The governing body should take advice from other stakeholders in the system, for example, through an advisory committee.In order to facilitate a robust program audit since the screened disorders are rare, programs may consider formal relationships between programs in neighboring jurisdictions. Such combined evaluation requires the harmonization of cutoffs and disorder definitions.

### 5.1. Biobanking

A clear policy is necessary defining the ownership and guardianship of residual neonatal screening specimens. The policy should define the following:Where and for how long residual specimens are stored and under what conditions.Whether and how and when residual specimens can be returned to families.Permitted uses of residual specimens and process and permissions required, e.g., for forensic use, research, laboratory quality activities, etc.

### 5.2. Neonatal Screening Data Access

A policy regarding access to data is necessary to ensure the privacy of the patient and the family is carefully protected.

Rigorous data security measures should be in place to protect all patient and family information and to ensure that services are resilient during periods of service disruption.

Access to both anonymized and identifiable data for research should be subject to the same requirements as access to bloodspots.

### 5.3. Coordination Between Services Related to Parts of Screening System

Collaboration between maternity services, the laboratory, and specialist pediatricians is necessary to develop a system that provides maximum support for families.

### 5.4. Program Monitoring/Audit/Governance

A screening program is complex, and evaluation must occur over several domains. Evaluation requires thoughtful data collection and analysis within and outside the laboratory. It must be a regular and ongoing process with a formal reporting structure. Reports should be available to all stakeholders and should be reviewed by a body with the authority to make adjustments to the program as needed. More frequent evaluation is necessary when changes have been made.

#### 5.4.1. Screening Pathway

Suggested monitoring includes

Age of baby at specimen collection;Transit time to laboratory;Laboratory time to reporting;Age of baby at time of referral to specialist;Number/% of unsuitable specimens.

The program should set performance targets (standards) and use these for quality improvement initiatives. Targets may be result- or disorder-specific, dependent on the clinical consequences of, e.g., delays in the availability of a screening result. They may be ideal, e.g., 100%, or achievable (<100%), and are typically set according to what is best for baby and achievable under the particular circumstances.

#### 5.4.2. Follow-Up of Positive Tests (Short-Term Follow-Up)

The suggested follow-up items (which may be specific to each disorder) are

Number/% with satisfactory follow-up in a time frame.Time/age of baby at diagnosis.Time/age of baby when treatment started.Health gains from screening (long term follow-up).Clinical outcome (was the disorder transient? did symptoms develop?) correlated to screening parameters, e.g., age at sampling, marker level/s, etc.

#### 5.4.3. Public Health Parameters

The suggested evaluation parameters are the following:Screening test, sensitivity;Screening test, specificity;Screening test, Positive Predictive Value;Investigation of false negative results.

Monitoring public health parameters informs the whole program. Where these are suboptimal, improvements may need to be made in all steps of the screening algorithms, i.e., the sample collection and timing, laboratory testing and cutoffs, second-tier testing, and the interface between the laboratory and the diagnostic and management clinicians.

Programs are encouraged to publish evaluation findings for local and international audiences, to inform case definitions, the natural history of disease (and elucidate relationships between laboratory findings and phenotype), and the effect of different treatments.

## 6. Consideration of Disorders to Be Added to Screening Panels

Programs should have a process whereby interested parties can apply to have disorders added to the screening panel and an appropriately multidisciplinary body to assess the applications.

The primary principles used to assess disorders should be the Wilson and Junger principles outlined above.

Some guidelines for neonatal screening have stipulated the need for evidence derived from high-quality randomized controlled trials showing benefits from presymptomatic diagnosis. Such evidence may not always be obtainable. Either there is already so strong a perception of benefit that trials will not be ethical (for example, with phenylketonuria or hypothyroidism) or very large numbers are required in each arm of the trial for relatively rare diseases and prolonged follow-up is required to demonstrate benefits from early detection and treatment.

Decisions should be based on published principles and/or criteria, the consideration procedures should be standardized and open to public scrutiny, and the result of deliberations should be published with the presented evidence and rationale for the decision.

When a disorder is being proposed for inclusion as part of a program, the following aspects merit consideration:Marker choice and validation of analytical methods.The use of prospective or retrospective samples, blinded or not.The development of a screening algorithm, including the confirmatory testing needed to fulfill a clear case definition.The need for a (limited) pilot study or evaluation to provide any evidence needed to inform decision making or to optimize practical aspects of pathway design.Family and healthcare professional information and education.

## 7. Genomics

The potential application of whole-genome sequencing (WGS) within NBS programs (both as a primary screen and a second- or higher-tier test) highlights a number of additional considerations for NBS programs. This area is complex and rapidly evolving and will be the subject of more detailed guidelines in the future.

The use of genomics within the context of neonatal screening is likely to require informed consent in most contexts. However, meaningful informed consent is difficult to achieve in this complex area, and programs need to consider how robust consent might be implemented at an individual and population level.

It is important for NBS programs to appreciate that the resulting information may have significant and potentially unwanted consequences for the wider family beyond the baby tested. This is not unique to genomics and does apply to other NBS tests as well. However, in the case of genomics, the resulting DNA profile/sequence may have lifelong implications for the baby. Programs should ensure that appropriate informed consent has been obtained to both the testing and retention of information; that there is capacity for secure storage of the resulting large amounts of data; and that the stored sequencing data should be re-analyzed if requested by a physician and after the consent of the parents or individual has been obtained. When developing a policy regarding the storage of genomic information, it should be noted that the evolution of sequencing technology and falling costs may allow for easier access to repeat sequencing when clinically indicated.

It may be that offline cold storage could cost-effectively store whole genomes for long periods of time. The advantage is that when, not if, there is a data breach and sequencing data are stolen from an NBS program, the impact would be reduced. Like other screening procedures that detect later-onset disorders, ethical issues such as the right not to know about adult-onset disorders or phenotypes must be considered.

When considering the use of genomic technology, programs should consider that currently, the phenotypic expression of genetic variants, whose significance has been established in individuals displaying clinical symptoms or with a known family history, is unclear in asymptomatic individuals making up the screened population. The significance of potentially important variants, prevalent in some ethnic groups, is poorly understood and consequently is not reported.

## 8. Ethics and Research

As stipulated in Section 5.1 and Section 5.2, there must be principles and criteria for access to residual specimens and data for research purposes accompanied by access request and decision-making processes.

Access would normally be subject to approval by the governance body conditional on appropriate ethics committee recommendations.

## 9. Summary

We realize that there is no single correct way to perform neonatal screening in a responsible way. Therefore, with these guidelines, we in no way want to be dogmatic. These guidelines are intended to indicate what elements of the screening process should be organized, not how they should be organized. As stated, the publication of these guidelines at this time offers a starting point for interactive discussion, and stakeholders are invited to critically review the guidelines and identify points for improvement.

## Data Availability

Data sharing is not applicable.

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
