# Peer review of "ISNS General Guidelines for Neonatal Bloodspot Screening 2025"

_2409-515X, 2025, doi:10.3390/ijns11020045_

Round 1
Reviewer 1 Report
Comments and Suggestions for Authors
Pls see attached file

Author Response
Reviewer 1
Comments and Suggestions for Authors
This manuscript presents a general guideline for the NBS based on DBS samples. The study provides anchors in both building and maintaining NBS programs. By the guidelines the authors aim to contribute to the enhancement of newborn screening protocols and intervention strategies. For healthcare professionals, especially in rare and not quite widespread specialties, like NBS or IEMs such guidelines are very important and of high importance. Moreover an interactive discussion in the field is highly welcome. Thank-you for your kind words.
Comments: Regarding the structure of the manuscript I would recommend to omit the paragraph 2 as title (since this is the title of the manuscript as all) and renumber according to the subparagraphs 2.x. and futher down accordingly. Eg: 2. Terminology; 3. Framework; 3. The screening pathway; 4. Neonatal screening system; 5. Expansion; 6. Ethics and research; 7. Genomic screening Thank-you – this is much tidier.
Suggestions: Line 44: It would be more informative if the number of babies were corrected to number/year. This number is per year – we have added the words ‘each year’.
Line 111: the terminology of screening pathway is presented in different format: I suggest to give it the same format and the short explanation – “the series of events…a baby” –the detailed explanations would be found in the according paragraph. Thank-you the screening pathway definition is presented separately because the other definitions are included in the CLSI Harmonised Terminology Database as described in the first sentence – this one is not,
Line 133: I suggest to omit the “(Principle 2)” from the paragraph title. In the paragraph body the referral to W&J Principle 2 could be made. This change has been made
Line 214: The indication of the web address for the list would be welcomed. Hence might be easier to reach. Alas this is not yet on the website, the sentence has been changed to read …. will be available ….
A “Summary” paragraph would be welcomed. Maybe lines 78-84 would serve as a good afterword. This is a good suggestion thank-you – the lines have been added as a part of a short summary.
Overall recommendation: In the manuscript a very important issue is discussed, when considering the implementation of a screening system/program in the healthcare. I consider it to be adequate for publication following a minor structural revision.

Reviewer 2 Report
Comments and Suggestions for Authors
This is an important guideline and the Reviewer thanks the authors for their work. Most of the Reviewer's comments are grammar and phrasing.
- Up until line 94 the document reads very informal. If this was the authors intention you can ignore this comment
- Line 21 the corresponding author information is missing
- Line 30 there is a dash after NBS
- Line 33 the sentence ends with a preposition
- Line 42, please remove one of the "is"
- Line 45, consider changing "two" to "more"
- Line 51-52, there are words missing after the abbreviation NGS
- Line 53, "posts" should be "poses"
- Line 63, could use the NBS abbreviation since it was previously defined
- Line 67, sentence ends with a preposition
- Line 79, here is an example of the informality the Reviewer mentioned
- Line 87, ends in 2 periods
- Line 93, could use NBS abbreviation
- Line 137, appears to be an extra space after "include"
- line 148, 167, 211, 286, etc in British English "eg" should be "e.g.", whereas in American English it is "e.g.,". The Reviewer does not care which the authors use, but please correct "eg" to an appropriate designation throughout the guideline
- Line 171, the second "of" should possibly be "for"
- Line 238, the Reviewer wants to confirm the authors meant to state "specimens not arriving"
- Line 243, 247, 312, 313, 314, etc follow-up, followup, and follow up, are all used interchangeable in the guideline. Please select one variation and be consistent
- line 263, if the authors choose the American English for "e.g.," then it should be "i.e.,". Otherwise ignore this comment
- line 346, the period is tabbed before genomics
- Line 352, the Reviewer wants to confirm roust was the intended word, because it did not seem to fit when reading. No changes are suggested if this was the intention
- In the genomics section it would be interesting to bring up the idea of using offline cold storage to cost-effectively store whole genomes for long periods of time. The advantage is that when, not if, there is data breach and sequencing data is stolen from a NBS program, the impact would be minimal compared to 23andMe.
Comments on the Quality of English Language
The document read well, the Reviewer only had minor suggestions to aid ESL readers and translation services.
Author Response
Reviewer 2
Thank-you for your thorough review, it is much appreciated.
- Up until line 94 the document reads very informal. If this was the authors intention you can ignore this comment Thank-you we have adjusted the language to be more formal.
- Line 21 the corresponding author information is missing Well spotted this has been added.
- Line 30 there is a dash after NBS Well spotted this has been removed.
- Line 33 the sentence ends with a preposition The ‘for’ has been deleted.
- Line 42, please remove one of the "is" Thank-you, fixed
- Line 45, consider changing "two" to "more" Change made thank-you this is better
- Line 51-52, there are words missing after the abbreviation NGS Thank-you this has been deleted at the suggestion of another reviewer.
- Line 53, "posts" should be "poses" As for the previous comment this has been deleted.
- Line 63, could use the NBS abbreviation since it was previously defined The abbreviation has been used, thank-you
- Line 67, sentence ends with a preposition Thank-you the sentence has been rewritten ‘It can serve as a checklist of items for which the screening program should have appropriate provisions’
- Line 79, here is an example of the informality the Reviewer mentioned Thank-you all this section has been rewritten in a formal style.
- Line 87, ends in 2 periods Thank-you one has been removed.
- Line 93, could use NBS abbreviation Thank-you as this refers to CLSI information we prefer to leave it as it is.
- Line 137, appears to be an extra space after "include" Thank-you the extra space has been removed.
- line 148, 167, 211, 286, etc in British English "eg" should be "e.g.", whereas in American English it is "e.g.,". The Reviewer does not care which the authors use, but please correct "eg" to an appropriate designation throughout the guideline Thank you- we have made changes according to the standard of the journal.
- Line 171, the second "of" should possibly be "for" Thank-you the sentence has been reworded ‘The screening pathway consists of the screening offer, sample …’
- Line 238, the Reviewer wants to confirm the authors meant to state "specimens not arriving" Yes thank-you this was intended
- Line 243, 247, 312, 313, 314, etc follow-up, followup, and follow up, are all used interchangeable in the guideline. Please select one variation and be consistent Thank-you all instances refer to the actions so the verb follow up is now used throughout.
- line 263, if the authors choose the American English for "e.g.," then it should be "i.e.,". Otherwise ignore this comment. We have now changed this as per your previous comment-thank you.
- line 346, the period is tabbed before genomics Thank-you this has been removed.
- Line 352, the Reviewer wants to confirm roust was the intended word, because it did not seem to fit when reading. No changes are suggested if this was the intention Thank-you for asking the work was meant to be robust and this has been changed.
- In the genomics section it would be interesting to bring up the idea of using offline cold storage to cost-effectively store whole genomes for long periods of time. The advantage is that when, not if, there is data breach and sequencing data is stolen from a NBS program, the impact would be minimal compared to 23andMe. Thank-you a paragraph has been added ‘It may be that offline cold storage could cost-effectively store whole genomes for long periods of time. The advantage is that when, not if, there is a data breach and sequencing data is stolen from a NBS program, the impact would be reduced.’

Reviewer 3 Report
Comments and Suggestions for Authors
Please share all attached comments with the authors.

Author Response
Reviewer 3
Thank-you for your thorough, comprehensive and knowledgeable review. We have made significant changes to the manuscript as detailed below.
ISNS Guideline review Abstract Lines 23-35: the readership would benefit from clarification that newborn screening encompasses three systems (blood spot, heart, and hearing screening) but only bloodspot screening is covered in this guideline. Thank-you – we have added the following sentence for clarification Although newborn screening encompasses testing for in the newborn period for critical congenital heart disease, hearing impairment, birth defects and congenital biochemical disorders (usually on bloodspots), this guideline is specifically about bloodspot screening.
Line 29: add “expand” as the guideline would also serve as a framework for the expansion of current programs. We agree, thank-you, sentence now includes expand to build, expand or maintain
Introduction: Line 42: the term “screened babies” is somewhat unclear from an epidemiologic definition; in light of global scrutiny of public health measures it would be also tremendously important to strengthen the statement regarding the overall success of NBS programs worldwide. Thank-you we have revised the sentence Neonatal screening or newborn screening (NBS) is a public health activity intended to serious treatable conditions shortly after birth. It has been recognized as an effective public health intervention by WHO. And will reference the Tam Lancet Global Health Nov 24 WHA77 statement paper.
Line 46: clarify “live normal lives because of timely treatment”, possibly include the concept of “timely disease management”. This statement should be improved. Thank-you we have replaced timely disease management with a new sentence Early detection and treatment enables achievement of maximum health and development potential in affected infants.
Line 51-55: this position is not neutral; NBS programs have seen and managed major technology disruption, for example with the introduction of MSMS platform technologies, have managed such challenges well and have – most importantly - emerged stronger after “disturbing [..] well-crafted programs”. As such, disturbances are common, should be considered as regularly occurring and a good program will find a new “steady-state”. Also, NGS-based technologies are already in use for select disorders as second-tier orthogonal methods. A more balanced presentation would benefit the reader. This is a good an valid point and we have removed most of that paragraph; the remainder is incorporated into the previous paragraph which now reads “Like other population screening programs, NBS has harms as well as benefits, and programs must always aim to do more benefit than harm (at a reasonable cost). Many, but not all, programs are built taking into account sets of ethical and practical rules to maintain programs that maximize health gain and avoid harm to the screened population. Not all developed programs have such rules and they are an essential part of introducing a new program. This prompted the ISNS to provide anchors for ethically and practically sound neonatal screening’
Line 116: replace with “The ISNS considers the Wilson and Junger criteria”… Thank-you we have deleted ‘so called’
Line 135: As a population wide initiative it is the individual baby but also the population at large. Please refine. We agree benefits to the baby benefit the wider population however it is arguable whether benefits to the wider population without benefits to the screened baby justify screening. We have modified the sentence ‘Accepted treatment (Principle 2) should be proven to result in reduced morbidity and mortality for the baby screened, the secondary consequence of this may also benefit the baby and the wider community.’
Line 148, 311: inconsistent use of “e.g.” Line 2.98 Program
We have repaired this throughout the document.
Monitoring/Audit/Governance This section title (2.4.4) would benefit from expanding and including “continuous process monitoring and program improvement”. In the document, NBS is in general described as a linear process (for example, referring to disrupting well-crafted programs”. In reality, NBS programs are never linear and really continuous processes with the before mentioned disruptions and expansions. Program monitoring assumes the important function of such “disruptions” and thereby leading to a new and better steady-state. Thank-you this is important. We have added two sentences . Added to the first paragraph ‘It must be a regular and ongoing process. Data should be reviewed by a body with the authority to make adjustments to the program as needed. More frequent evaluation is necessary when changes have been made.’ And after the first set of bullets ‘They may be Gold (ideal) eg 100%, or achievable (<100%) and are typically set compromising best for baby with achievable in the particular circumstances’ We consider under Public Health parameters Positive predictive value covers the % of affected cases /screen positive babies.
Please consider including in the list of monitoring the concept of screen positive cases in the context of the percentage of truly affected cases and opportunities to refine screening algorithms in the interphase between the laboratory and clinical management. We have added a sentence under public health parameters ‘Monitoring public health parameters informs the whole program. Where these are suboptimal, improvements may need to be made in all steps of the screening algorithms ie sample collection and timing, laboratory testing and cutoffs, second tier testing, and the interface between the laboratory and the diagnostic and management clinicians.’
Line 309-311: strengthen this paragraph, remove “ideally”. A program without targets is generally not accountable and cannot improve. Thank-you this has been done.
Line 317, 318 context of “developed for each disorder” is unclear We agree this is unclear and have moved the concept to the introductory sentence Suggested follow up items (which may be specific to each disorder) are:
Line 327-330: An important missing component and opportunity is the concept of regularly reporting program performance to all stakeholders. This not only generates accountability but provides a continuous “use case and need” for NBS and expanding thereof. We should not forget Ellen Gablers work that really triggered wide-spread awareness and process and program transparency measures and improvements. This is a good point and the following sentence has been added to the introductory paragraph in the section ‘It must be a regular and ongoing process with a formal reporting structure. Reports should be available to all stakeholders and should be reviewed by a body with the authority to make adjustments to the program as needed. More frequent evaluation is necessary when changes have been made.’
Line 337-342 “and the prolonged follow—up required” is unclear. Thank-you the following has been added to the sentence ;required to demonstrate benefits from early detection and treatment.’
The discussion would benefit from also including the current nonsensical requirement of pilot studies requiring the prospective identification of an affected baby as such practices only delay panel expansion and add objective evidence value.
The ISNS recognizes the value of well organised pilot studies and evaluations designed to provide evidence that may be needed to help national decision making eg the prevalence of a condition in the population where screening is planned. In addition these studies may help optimise the design and logistics of screening prior to national implementation. The ISNS in these guidelines firmly stands by that approach. To meet this reviewers point, we have elaborated on what items can be considered prior to the introduction of a new disorder and we hope that satisfies this reviewer.
the following has been added
When a disorder is being proposed for inclusion as part of a programme, the following aspects merit consideration:
- Marker choice and the validation of analytical methods
- The use of prospective or retrospective samples, blinded or not
- The development of a screening algorithm including the confirmatory testing needed to fulfill a clear case definition
- The need for a (limited) pilot study or evaluation to provide any evidence needed to inform decision making or to optimize practical aspects of pathway design.
- Family and healthcare professional information and education
Line 346 The section “Genomics” is not balanced and would deserve a more thorough presentation. The consequences of genomic testing stated for “a wider family beyond the baby tested” is even without genomic tools a reality in NBS today (e.g. XALD screening). Variant analysis as orthogonal methodology is frequently in use without additional consent. Considering this unbalanced perspective, this section could be condensed and the reader referred to future discussion of this subject. Thank-you this is indeed complex and screening using metabolic markers also has wider family implications as you suggest and this is implied in the discussion. More detailed guidance is required as you suggest (this will be the first task for the newly formed ISNS Genomics Committee) and the following sentence has been added ‘This area is complex and rapidly evolving and will be the subject of more detailed guidelines in future.’

Round 2
Reviewer 3 Report
Comments and Suggestions for Authors
Thank you to the authors for the improvements and considering the suggestions.